# Timing-Dependent Effects of Transcranial Direct Current Stimulation on Hand Motor Function in Healthy Individuals: A Randomized Controlled Study

**DOI:** 10.3390/brainsci11101325

**Published:** 2021-10-06

**Authors:** Nam-Gyu Jo, Gi-Wook Kim, Yu Hui Won, Sung-Hee Park, Jeong-Hwan Seo, Myoung-Hwan Ko

**Affiliations:** 1Department of Physical Medicine & Rehabilitation, Jeonbuk National University Medical School, Geonjiro 20, Deokjin-gu, Jeonju 54097, Korea; cnk9016@jbnu.ac.kr (N.-G.J.); k26@jbnu.ac.kr (G.-W.K.); wonyh@jbnu.ac.kr (Y.H.W.); shpark0130@jbnu.ac.kr (S.-H.P.); vivaseo@jbnu.ac.kr (J.-H.S.); 2Research Institute of Clinical Medicine of Jeonbuk National University-Biomedical Research Institute of Jeonbuk National University Hospital, Geonjiro 20, Deokjin-gu, Jeonju 54907, Korea

**Keywords:** timing-dependent effect, transcranial direct current stimulation, cortical excitability, neuroplasticity

## Abstract

The timing of transcranial direct current stimulation (tDCS) is essential for enhancing motor skill learning. Previously, tDCS, before or concurrently, with motor training was evaluated in healthy volunteers or elderly patients, but the optimal timing of stimulation has not been determined. In this study, we aimed to optimize the existing tDCS protocols by exploring the timing-dependent stimulation effects on finger movements in healthy individuals. We conducted a single-center, prospective, randomized controlled trial. The study participants (n = 39) were randomly assigned into three groups: tDCS concurrently with finger tapping training (CON), tDCS prior to finger tapping training (PRI), and SHAM-tDCS simultaneously with finger tapping training (SHAM). In all groups, the subjects participated in five 40-min training sessions for one week. Motor performance was measured before and after treatment using the finger-tapping task (FTT), the grooved pegboard test (GPT), and hand strength tests. tDCS treatment prior to finger tapping training significantly improved motor skill learning, as indicated by the GPT and hand strength measurements. In all groups, the treatment improved the FTT performance. Our results indicate that applying tDCS before training could be optimal for enhancing motor skill learning. Further research is required to confirm these findings.

## 1. Introduction

Non-invasive brain stimulation (NIBS) is commonly used to study brain physiology and to modulate brain activity. The two most commonly used techniques for NIBS are transcranial direct current stimulation (tDCS) and transcranial magnetic stimulation (TMS) [1,2,3]. NIBS modulates the excitability of the stimulated cortex depending of the stimulation protocol, and has been used in various fields for brain physiological research [4]. It is known that NIBS application to the primary motor cortex (M1) alters neuroplasticity and is important for motor performance [5,6]. A recent tDCS study revealed evidence for the involvement of the dorsolateral prefrontal cortex (DLPFC) in human cue-guided choice [7]. Another study of TMS stimulation over DLPFC found that it interfered with fear-memory consolidation and reduced return of fear [8]. The therapeutic potential of NIBS is also recognized in cognitive and speech therapy, physical rehabilitation, and psychiatry [9,10,11,12].

tDCS induces brain polarization by applying a weak current to the scalp; the anodal polarity increases cortical excitability, while cathodal polarity decreases it [5,6,13,14]. tDCS is widely used for post-stroke rehabilitation [11,12,15]. In healthy individuals, anodal tDCS improves motor function, reaction times, and motor skill learning by enhancing cortical excitability [16,17,18].

To facilitate motor training with tDCS, it is important to consider the timing of stimulation [19,20]. Although several studies compared tDCS stimulation before or concurrently with motor training in healthy volunteers or the elderly, the optimal timing has not been established [19]. Research involving stroke patients and healthy volunteers reported that tDCS should be applied prior to motor training [21,22], but opposing results indicate that using tDCS concurrently with motor training is more effective [23].

Learning a sequential motor involves integrating separate movements into a unified and coordinated sequence of actions through practice [24,25]. Motor sequence learning implies enhancement of synaptic connectivity, and motor inhibition and cognitive ability such as working memory are involved [26,27,28,29]. Brain areas such as the primary motor cortex (M1) and lateral prefrontal cortex are engaged in motor learning depending on the task [7,30]. A sequential finger-tapping task (FTT) is commonly used to evaluate motor-sequence learning by measuring the speed and accuracy of entering a presented number with the corresponding finger [31].

The dependence of tDCS outcomes on the timing of application has not been fully explored [20]. Understanding the time-dependent effects of tDCS in conjunction with physical training is crucial for neurological rehabilitation. In this study, we analyzed the time-dependent effects of tDCS on motor skill learning in healthy individuals.

## 2. Materials and Methods

### 2.1. Participants

The study participants were recruited through a notice posted on the hospital’s bulletin board and screened by a rehabilitation physician. Written informed consent was obtained from all the participants prior to enrollment, and all research procedures were conducted in accordance with the ethical principles of the Declaration of Helsinki.

Healthy individuals aged 20–85 years were eligible for participation if no clinically significant diseases or contraindications to tDCS were detected. All subjects understood the research purpose and procedures and confirmed their willingness to participate in clinical research through voluntary consent. This study was approved by the Institutional Review Board of Jeonbuk National University Hospital (approval number: CUH 2019-12-036).

### 2.2. Trial Design

This study was a single-center, prospective, randomized controlled trial conducted at the rehabilitation center of Jeonbuk National University Hospital. The recruited individuals were randomly assigned to one of the three groups according to the study protocol: (1) concurrent tDCS with finger tapping training (CON group); (2) tDCS applied prior to finger tapping training (PRI group); (3) SHAM-tDCS applied simultaneously with finger tapping training (SHAM group). A clinical research coordinator, a nurse who was not involved in the assessments, was responsible for the randomization. Measurements were conducted at two time points: evaluation 1 (E1, pre-treatment) and evaluation 2 (E2, just after treatment). All investigators remained blinded during data acquisition and analysis.

The study was registered at the Clinical Research Information Service, under the direction of the Korea Centers for Disease Control and Prevention (registration number: KCT0005838).

### 2.3. Experimental Setup

All subjects participated in five 40-min training sessions for one week, one session per day. The flow chart of interventions is shown in Figure 1. The CON group received 20 min of finger-tapping training with the non-dominant hand concurrently with anodal tDCS applied to the primary motor cortex (M1) on the non-dominant hemisphere. The PRI group received 20 min of anodal M1 tDCS, followed by 20 min of finger-tapping training. The SHAM group received 20 min of finger-tapping training concurrently with SHAM-tDCS over the M1. During the finger tapping training, the subjects were instructed to press the keyboard key when presented a random number from one to four on a computer screen. The presented numbers corresponded to the fingers of the non-dominant hand (e.g., the index finger for number one, the middle finger for number two, etc.). All training courses were guided and supervised by a physician to ensure that all participants exercised faithfully.

### 2.4. tDCS Protocol

A direct current stimulator (NeuroConn Ltd., Ilmenau, Germany) was used to deliver anodal tDCS. The anode electrode was placed over the motor hotspot of the first dorsal interosseous (FDI) on the non-dominant hemisphere; the cathode electrode was placed on the contralateral supraorbital area (Figure 2). The duration of tDCS was 20 min. The electrode size was 35 cm^2^, and the stimulation intensity was 2 mA, in accordance with the current safety limitations [32]. The tDCS protocol (2 mA of anodal tDCS for 20 min) was selected based on the existing evidence [33]. During the SHAM conditions, a current flowed for a period of 30 s at the beginning of stimulation and then turned off [34].

### 2.5. Motor Performance Measurements

Motor performance was measured using the finger tapping task (FTT), the grooved pegboard test (GPT), and hand strength tests (grip power, lateral, palmar, and tip pinches).

The FTT is a finger tapping task, in which a total of 120 random numbers were entered using the dominant or non-dominant hand, as described above. Based on the FTT, accuracy and reaction time were measured and converted into the skill parameter using the following equation: Skill = accuracy/mean reaction time [33].

The GPT requires the insertion of 25 pegs into the holes as quickly as possible. The grooved pegboard apparatus consisted of a metal surface (10.1 × 10.1 cm^2^) with a 5 × 5 matrix of keyhole-shaped holes in various orientations. Each peg (3 mm in diameter) had a small ridge on one side. The subject was instructed to insert pegs using only one hand in a specific order, demonstrated by the examiner. The subject was asked to work with the right hand from left to right, and with the left hand in the opposite direction. Each trial consisted of the task performed with the dominant hand first and repeated immediately with the non-dominant hand. The subjects collected only one peg at a time, and if a peg was dropped, a new peg was drawn from the pile. The score for each hand was the time required to complete the task, and the timing was not interrupted in the event of a dropped peg. Each subject performed two trials, and the fastest time was recorded [35].

A hand dynamometer (JAMAR^®^, Chicago, IL, USA) was used to measure the maximum grip strength (Kg) in elbow flexion and shoulder abduction position [36]. Pinch strength was measured for the three pinch types: tip, palmar, and lateral pinches. The tip pinch was measured by gripping the objects with the tip of the index finger and the thumb. This parameter indicates the direct strength of the two fingers. The palmar pinch was measured by gripping the objects with the thumb, the index finger, and the middle finger. The lateral pinch was measured by gripping the objects with the thumb and the lateral side of the index finger. The levers of the dynamometer and pinch gauge were same width during each test. Therefore, the subject could start gripping or pinching with the same force using the same muscles [37]. The highest value was obtained by measuring the hand strength twice in each position.

Safety was assessed by monitoring adverse reactions, such as subjective awareness or symptoms that were self-reported by the participants.

### 2.6. Statistical Analyses

Statistical analyses were performed using the SPSS software (version 23.0; IBM, Armonk, NY, USA). Data were presented as means (standard deviations, SD) for continuous variables and frequencies for categorical variables. For continuous variables, differences in baseline characteristics were evaluated with one-way analysis of variance (ANOVA) if the normality criteria were met. Otherwise, the Kruskal-Wallis test was used. The Pearson chi-squared test was used to compare the differences in demographic characteristics between groups for categorical variables. The intra-group analysis of outcome data was performed using the paired *t*-test or Wilcoxon signed-rank test based on the normality test results. Inter-group analysis was conducted using one-way ANOVA or Kruskal-Wallis tests based on the normality test results, followed by planned multiple pairwise comparisons with Bonferroni correction. The alpha level was set to 0.05, and the *p* values less than 0.05 indicated significant differences.

## 3. Results

### 3.1. Baseline Measurement

Thirty-nine participants (13 in each group) were recruited between 15 June and 13 November 2020. Except for one participant who withdrew consent, 38 participants completed the treatment and both measurements (Figure 3).

The baseline demographic characteristics and the results of motor task measurements are shown in Table 1. Overall, 34 participants were right-handed, and four were left-handed. There were no significant differences in baseline demographic characteristics and task performance between the three groups.

### 3.2. Finger Tapping Task

The one-way ANOVA analysis of the combined Skill values revealed no significant differences between the three groups (*p* = 0.498). Intra-group comparisons indicated a significant increase in the Skill score in all groups after the treatment (*p* < 0.001 in the CON and PRI groups, and *p* = 0.001 in the SHAM group, Table 2).

### 3.3. Grooved Pegboard Test

The GPT measurements indicated motor improvement after treatment in both CON and PRI groups (time to accomplish the task decreased). The observed differences were significant for the PRI group only (*p* = 0.006, Figure 4). The SHAM group showed an increased execution time (*p* = 0.050). The inter-group comparisons indicated that the CON and PRI groups were significantly different from the SHAM group.

### 3.4. Hand Strength

Although the one-way ANOVA analysis of hand strength revealed no significant differences between the groups, the intra-group analysis showed a significant increase in the lateral and palmar pinch strength for the PRI group (*p* = 0.027 and *p* = 0.003 for the lateral and palmar pinch, respectively, Figure 4).

No adverse effects related to the applied treatments were observed in this study.

## 4. Discussion

Our results indicate that tDCS before motor training induces a significant improvement in the GPT performance and increases the lateral and palmar pinch strength. In addition, GPT revealed significant differences between the three groups. All groups showed a significant improvement in the Skill value of the FTT. It was observed that the tDCS effects on fine motor function and pinch strength are timing-dependent, and the application of tDCS before training is more advantageous than concurrent application.

The timing of tDCS application is an important factor to consider in the tDCS protocol for motor training [20]. Several studies have compared tDCS stimulation before or concurrently with motor training in healthy volunteers or the elderly. However, the evidence for the timing-dependent effect of tDCS is still insufficient to warrant a recommendation, thus requiring further research.

Our findings are consistent with previous studies. In a recent randomized controlled trial, which explored the time-dependent effects of tDCS on upper limb function using mirror therapy in patients with chronic stroke, tDCS before training improved daily function and kinematics compared with the concurrent stimulation [21]. In other studies investigating the time-dependent effects in healthy participants, the cortical excitability was enhanced when tDCS was applied before motor training [22].

tDCS stimulation provokes changes in the resting membrane potential of the cortical interneurons. Anodal stimulation increases neural excitability and increases neuronal firing rate [38]. The underlying neurophysiological mechanisms are complexly regulated on the cellular level and in brain networks [39]. Additionally, the long-term effect of tDCS is related to inhibitory interneurons expressing γ-aminobutyric acid (GABA), which affect motor learning and motor memory processing [40,41,42]. Excitatory transcranial brain stimulation enhances activity-dependent plasticity and is known as gating [43,44]. After stimulation, the lasting effect of tDCS evolves through modulation of the N-methyl-D-aspartate (NMDA) receptor-dependent glutathione interneurons [14]. Synaptic plasticity manifests as long-term potentiation (LTP), depression (LTD), or bidirectional plasticity. Based on the homeostatic plasticity model, synaptic transmission can be facilitated or impeded depending on the dynamic modification thresholds which arise from the adaptation of post-synaptic activity [45,46]. Corticospinal excitability during motor activity can be primed by tDCS, inducing neuroplastic effects [47]. Recent studies have shown that the application of tDCS prior to activity execution improves motor performance stronger than concurrent application [48]. This effect can be explained by the interference between simultaneous tDCS and excitatory effects from motor activity due to regulatory homeostatic mechanisms. Therefore, tDCS should be applied prior to motor activity to avoid homeostatic regulation [22,49].

In our study, the Skill values obtained from the FTT measurements indicated significant motor improvements in all groups, which potentially is a result of learning during training. It is known that tDCS has minimal impact on simple reaction time tasks, and the FTT improvements observed in this study were likely associated with the learning effect [20].

Motor skill learning was significantly enhanced in the PRI group, as indicated by the GPT. This test is most frequently used for assessing fine motor function [50] and is closely related to daily life activities [51]. A recent study reported a time-dependent effect in stroke patients. The group that received tDCS before motor training exhibited better recovery of daily functions than the group with simultaneous tDCS application [21]. In another study involving stroke patients, the 20-min tDCS before physical exercise increased muscle strength and reduced upper limb motor dysfunction [52].

In this study, the application of tDCS before motor training significantly increased hand grip strength. Previously, the grip force during upper limb exercise was significantly increased in the experimental group receiving tDCS before the test, which is similar to our results [52]. We applied anodal tDCS at the hot spot of the FDI muscle, and thus it could be expected that the pinch strength would increase by a greater extent compared to the grip strength.

This study was subject to several limitations. First, the blinding of participants may have been limited by the different treatment durations between the groups and the SHAM stimulation was only applied during training and not before training. Second, this study included a small number of participants and was conducted at a single center. These limitations should be taken into account in future studies. 

## 5. Conclusions

When comparing the during-effect and the lasting-effect of tDCS, the lasting effect was more evident in fine motor skills and pinch strength. Our findings suggest that the time shortly before motor training could be optimal for tDCS application. Further research is required to confirm these findings.

## Figures and Tables

**Figure 1 brainsci-11-01325-f001:**
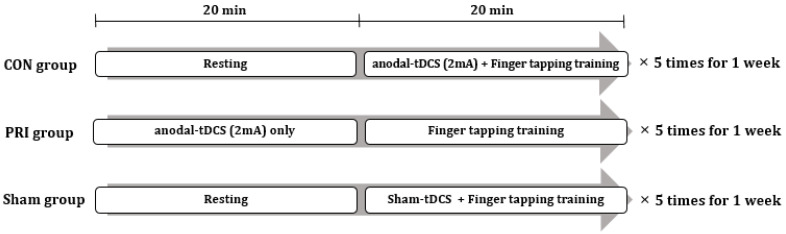
Experimental design involving the three intervention groups. tDCS: anodal transcranial direct current stimulation.

**Figure 2 brainsci-11-01325-f002:**
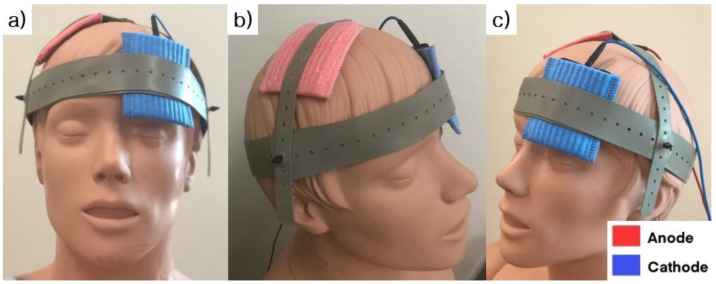
The location of the stimulating electrodes used for transcranial direct current stimulation. (**a**) right oblique view, (**b**) front view and (**c**) left oblique view are presented. The anode electrode (Red) was placed over the motor hotspot of the first dorsal interosseous (FDI) on the non-dominant hemisphere; the cathode electrode (Blue) was placed on the contralateral supraorbital area.

**Figure 3 brainsci-11-01325-f003:**
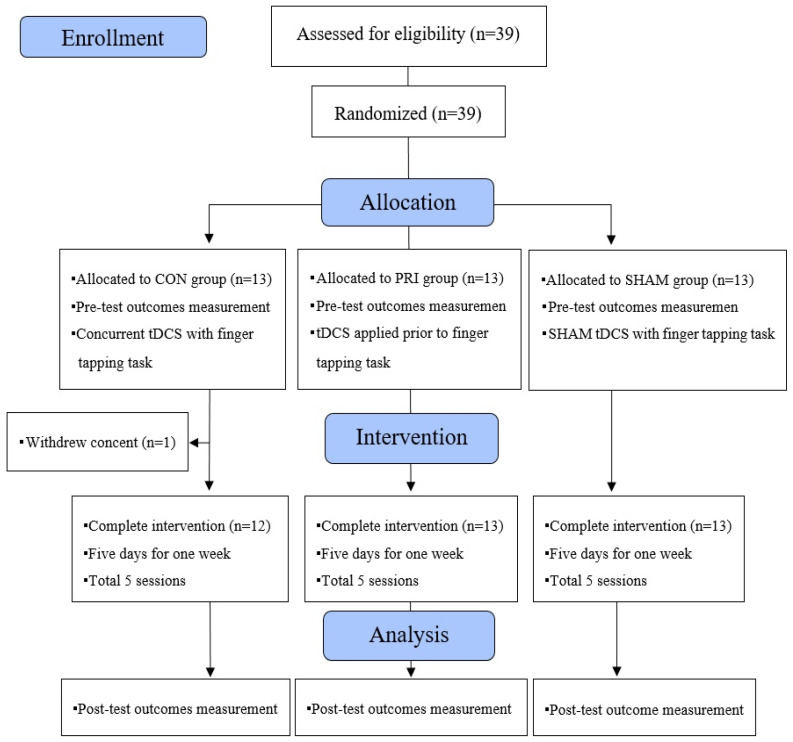
The CONSORT flow diagram showing experimental pipeline.

**Figure 4 brainsci-11-01325-f004:**
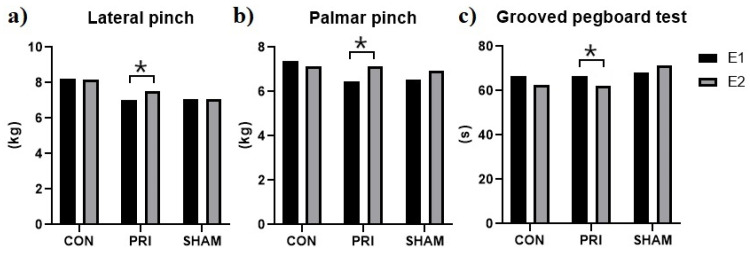
Motor performance assessment. (**a**) Lateral pinch, (**b**) Palmar pinch, (**c**) Grooved pegboard test. Only the PRI group showed significant improvement in the lateral and palmar pinch and the grooved pegboard tests. * Significant differences were determined using paired *t*-tests.

**Table 1 brainsci-11-01325-t001:** Comparison of the baseline demographic characteristics and motor task performance between the three groups. Data are presented as frequencies (%) and means (standard deviations, SD).

	CON (n = 12)	PRI (n = 13)	SHAM (n = 13)	*p*-Value
Age (years)	27.58 (9.12)	25.69 (4.13)	28.23 (12.54)	0.820 ^2^
Sex				
Male	9 (23.7%)	7 (18.4%)	7 (18.4%)	0.464 ^3^
Female	3 (7.9%)	6 (15.8%)	6 (15.8%)	
Non-dominant hand				
Right	0 (0.0%)	1 (2.6%)	3 (7.9%)	0.157 ^3^
Left	12 (31.6%)	12 (31.6%)	10 (26.3%)	
Finger tapping task				
Skill	0.15 (0.02)	0.16 (0.01)	0.15 (0.03)	0.339 ^1^
Grooved pegboard test				
Time (s)	66.61 (7.10)	66.78 (7.83)	68.11 (13.13)	1.000 ^2^
Hand strength (Kg)				
Grip power	37.28 (10.34)	30.52 (11.00)	27.37 (9.97)	0.068 ^1^
Lateral pinch	8.24 (1.97)	7.01 (1.83)	7.05 (2.07)	0.220 ^1^
Palmar pinch	7.39 (1.93)	6.44 (1.04)	6.56 (1.95)	0.323 ^1^
Tip pinch	5.26 (1.63)	5.03 (1.00)	5.42 (1.89)	0.812 ^1^

^1^ One-way ANOVA, ^2^ Kruskal-Wallis test, ^3^ Chi-square test.

**Table 2 brainsci-11-01325-t002:** Motor task performance. Data are presented as frequencies (%) and means (standard deviations, SD).

Variables	Group	E1	E2	Intra-Group *p*-Value	Inter-Group *p*-Value
Finger tapping task (Skill)	CON	0.15 (0.02)	0.20 (0.03)	<0.001 ^1^	0.498 ^3^
	PRI	0.16 (0.01)	0.22 (0.02)	<0.001 ^1^	
	SHAM	0.15 (0.03)	0.20 (0.04)	0.001 ^2^	
Grooved pegboard test (s)	CON	66.61 (7.10)	62.71 (4.85)	0.059 ^1^	0.002 ^3^
	PRI	66.78 (7.83)	62.36 (7.69)	0.006 ^1^	
	SHAM	68.11 (13.13)	71.59 (11.79)	0.050 ^1^	
Grip power (Kg)	CON	37.28 (10.34)	35.96 (11.17)	0.173 ^1^	0.511 ^3^
	PRI	30.52 (11.00)	30.50 (11.78)	0.970 ^1^	
	SHAM	27.37 (9.97)	27.03 (10.22)	0.708 ^1^	
Lateral pinch (Kg)	CON	8.24 (1.97)	8.20 (2.04)	0.839 ^1^	0.156 ^3^
	PRI	7.01 (1.83)	7.54 (1.77)	0.027 ^1^	
	SHAM	7.05 (2.07)	7.09 (1.72)	0.854 ^1^	
Palmar pinch (Kg)	CON	7.39 (1.93)	7.13 (2.43)	0.306 ^2^	0.059 ^3^
	PRI	6.44 (1.04)	7.14 (1.36)	0.003 ^1^	
	SHAM	6.56 (1.95)	6.95 (2.07)	0.166 ^1^	
Tip pinch (Kg)	CON	5.26 (1.63)	5.49 (1.94)	0.341 ^1^	0.222 ^3^
	PRI	5.03 (1.00)	5.60 (1.35)	0.093 ^1^	
	SHAM	5.42 (1.89)	5.28 (1.53)	0.641 ^1^	

^1^ Paired *t*-test, ^2^ Wilcoxon signed-rank test for the difference, ^3^ One-way ANOVA. Skill = accuracy/mean reaction time.

## Data Availability

The accompanying data are fully presented as graphs and tables. Raw data can be obtained upon reasonable request from the corresponding author.

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
