# Peer review of "Timing-Dependent Effects of Transcranial Direct Current Stimulation on Hand Motor Function in Healthy Individuals: A Randomized Controlled Study"

_brainsci, 2021, doi:10.3390/brainsci11101325_

Round 1
Reviewer 1 Report
In the present study entitled ‘Timing-dependent Effects of Transcranial Direct Current Stimulation on Hand Motor Function in Healthy Individuals: A Randomized Controlled Study’, by Jo and colleagues, authors aimed to examine timing-dependent transcranial direct current stimulation (tDCS) effects on finger movements. For this purpose, 39 participants were randomly divided into three groups, based on stimulation protocol: concurrent tDCS with finger tapping training group, tDCS applied prior to finger tapping training group and sham-tDCS applied simultaneously with finger tapping training group. Results showed tDCS stimulation prior to motor training significantly improves; moreover, the stimulation significantly improves performance the finger-tapping task’s performance.
In general, I think the idea of the study is interesting and the intriguing results that the authors have found might be of interest to the readers of Brain Sciences. However, some minor comments, as well as some crucial citations that should be included to support the authors’ claim, need to be addressed to improve the research article and its readability prior the publication in the present form.
Comments:
- Regarding the abstract: according to the Journal’s guidelines, authors should have provided an abstract written as a single paragraph, without headings, of about 200 words maximum. Indeed, the current one includes 212 words.
- Page 1, line 36: correct the typo ‘time’ with ‘times’.
- Page 1-2, Introduction: The introduction is well written and structured. However, in my opinion, a general overview of non-invasive brain stimulation (NIBS) commonly used to modulate brain activity and how they are applied to alter various cognitive domains is needed for non-expert readers. Thus, I would suggest some references that would be crucial in this section and methodologically fit with the present manuscript: for example, Garofalo and colleagues (2021, Cortex) delivered transcranial direct current stimulation (tDCS) to the dorsolateral prefrontal cortex (DLPFC) and show causal evidence for the involvement of DLPFC in two forms of human cue-guided tasks, having crucial implication in addiction and dependence. Also, I would suggest to cite results from Borgomaneri and colleagues’ recent study (2020, Current Biology) that show causal evidence for the application of rTMS over DLPFC after memory reactivation in reducing responding to learned fear. The same research group (Borgomaneri et al., 2021, Brain sciences) provided interesting insights on the use of single-pulse TMS in the modulation of corticospinal excitability of the right and the left M1 during the processing of emotional facial expressions. Finally, I would suggest Borgomaneri and colleagues’ study (2021, Journal of Affective Disorders), that illustrated the therapeutic potential of NIBS as a valid alternative for those patients not responding to psychotherapy and/or drug treatments. I believe that adding information regarding different applications of NIBS will dramatically improve the flow.
- Page 2, lines 43-44: Authors stated that ‘Learning a sequential motor behavior involves...coordinated sequence of actions through practice’. In this regard, I would suggest adding some key references that could provide further insight on how instrumental motor performances can be promoted by associative learning mechanisms. Evidence from Garofalo and colleagues’ study (2019, Scientific Reports), in which 100 healthy participants performed a decision-making and learning task, show how individual differences in cognitive abilities are critically involved in the ability of Pavlovian stimuli to promote associated instrumental motor responses previously associated with a reward (i.e., PIT effect). Also, results from a recent study by the same research group (Garofalo et al., 2021, Cortex) show causal evidence for the involvement of the DLPFC in the outcome representation needed to guide cue-guided motor actions. Adding to this evidence, in a recent review, Borgomaneri and colleagues (2020, Cortex) outlined neural circuits of motor inhibition in humans; the same research group (Battaglia et al., 2021, Behaviour Research and Therapy) also explored the impact that emotional stimuli have on motor inhibition of ongoing learned actions.
- Page 2, lines 80-81: authors should be more specific and indicate if anodal tDCS was applied to the ipsilateral motor cortex (as the acronym ‘iM1’ suggests).
- Regarding the figures: I suggest adding a figure that displays stimulation sites’ position on the scalp.
- Page 8-9, Bibliography: authors should provide the abbreviated journal name in italics to the following citations Hummel et al., 2006; Elsner et al., 2019; Elsner et al., 2017; Antal et al., 2004; Keele et al., 2003; Doyon et al., 2018; Saucedo Marquez et al., 2013; Rahman et al., 2013
- Page 9, lines 326-327: correct the authors’ name and order of citation to the following citations ‘CAUMO, W.; Souza, I.C.; Torres, I.L.; Medeiros, L.; Souza, A.; Deitos, A.; Vidor, L.; Fregni, F.; Volz, M.S.J.F.i.p. Neurobiological effects of transcranial direct current stimulation: a review. 2012, 3, 110. DOI:10.3389/fpsyt.2012.00110’.
Author Response
[Reviewer 1]
Comments and Suggestions for Authors
- In the present study entitled ‘Timing-dependent Effects of Transcranial Direct Current Stimulation on Hand Motor Function in Healthy Individuals: A Randomized Controlled Study’, by Jo and colleagues, authors aimed to examine timing-dependent transcranial direct current stimulation (tDCS) effects on finger movements. For this purpose, 39 participants were randomly divided into three groups, based on stimulation protocol: concurrent tDCS with finger tapping training group, tDCS applied prior to finger tapping training group and sham-tDCS applied simultaneously with finger tapping training group. Results showed tDCS stimulation prior to motor training significantly improves; moreover, the stimulation significantly improves performance the finger-tapping task’s performance.
- In general, I think the idea of the study is interesting and the intriguing results that the authors have found might be of interest to the readers of Brain Sciences. However, some minor comments, as well as some crucial citations that should be included to support the authors’ claim, need to be addressed to improve the research article and its readability prior the publication in the present form.
Comments:
- Regarding the abstract: according to the Journal’s guidelines, authors should have provided an abstract written as a single paragraph, without headings, of about 200 words maximum. Indeed, the current one includes 212 words.
- Response : The heading of abstract was deleted, and the abstract was corrected to 197 words.
- Page 1, line 36: correct the typo ‘time’ with ‘times’.
- Response : This word has been corrected to ‘times’. (Page 2, line 4)
- Page 1-2, Introduction: The introduction is well written and structured. However, in my opinion, a general overview of non-invasive brain stimulation (NIBS) commonly used to modulate brain activity and how they are applied to alter various cognitive domains is needed for non-expert readers. Thus, I would suggest some references that would be crucial in this section and methodologically fit with the present manuscript: for example, Garofalo and colleagues (2021, Cortex) delivered transcranial direct current stimulation (tDCS) to the dorsolateral prefrontal cortex (DLPFC) and show causal evidence for the involvement of DLPFC in two forms of human cue-guided tasks, having crucial implication in addiction and dependence. Also, I would suggest to cite results from Borgomaneri and colleagues’ recent study (2020, Current Biology) that show causal evidence for the application of rTMS over DLPFC after memory reactivation in reducing responding to learned fear. The same research group (Borgomaneri et al., 2021, Brain sciences) provided interesting insights on the use of single-pulse TMS in the modulation of corticospinal excitability of the right and the left M1 during the processing of emotional facial expressions. Finally, I would suggest Borgomaneri and colleagues’ study (2021, Journal of Affective Disorders), that illustrated the therapeutic potential of NIBS as a valid alternative for those patients not responding to psychotherapy and/or drug treatments. I believe that adding information regarding different applications of NIBS will dramatically improve the flow.
- Response : Thanks for your suggestion. An introduction to a general overview of NIBS, including your recommended references was added. (Page 1, line 32-43)
- Added paragraph (Page 1, line 32-43) : Non-invasive brain stimulation (NIBS) is commonly used to study brain physiology and to modulate brain activity. The two most commonly used techniques for NIBS are Transcranial direct current stimulation (tDCS) and transcranial magnetic stimulation (TMS) [1-3]. NIBS modulates the excitability of the stimulated cortex depending of the stimulation protocol, is has been used in various fields for brain physiological research [4]. It is known that NIBS application to primary motor cortex (M1) alters neuroplasticity and is important for motor performance [5,6]. A recent tDCS study revealed evidence for the involvement of the dorsolateral prefrontal cortex (DLPFC) in human cue-guided choice [7], Another study of TMS stimulation over DLPFC found that it interfered with fear-memory consolidation and reduced return of fear [8]. The therapeutic potential of NIBS is also recognized in cognitive and speech therapy, physical rehabilitation, and psychiatry [9-12].
- Page 2, lines 43-44: Authors stated that ‘Learning a sequential motor behavior involves...coordinated sequence of actions through practice’. In this regard, I would suggest adding some key references that could provide further insight on how instrumental motor performances can be promoted by associative learning mechanisms. Evidence from Garofalo and colleagues’ study (2019, Scientific Reports), in which 100 healthy participants performed a decision-making and learning task, show how individual differences in cognitive abilities are critically involved in the ability of Pavlovian stimuli to promote associated instrumental motor responses previously associated with a reward (i.e., PIT effect). Also, results from a recent study by the same research group (Garofalo et al., 2021, Cortex) show causal evidence for the involvement of the DLPFC in the outcome representation needed to guide cue-guided motor actions. Adding to this evidence, in a recent review, Borgomaneri and colleagues (2020, Cortex) outlined neural circuits of motor inhibition in humans; the same research group (Battaglia et al., 2021, Behaviour Research and Therapy) also explored the impact that emotional stimuli have on motor inhibition of ongoing learned actions.
- Response : Thanks for recommending a good reference. Additional explanations about motor learning were further described. (Page 2, line 13-17)
- Revised paragraph (added sentences are underlined, Page 2, line 13-17) : Learning a sequential motor involves integrating separate movements into a unified and coordinated sequence of actions through practice [24,25]. Motor sequence learning implies enhancement of synaptic connectivity, and motor inhibition and cognitive ability such as working memory are involved [26-29]. Brain areas such as primary motor cortex (M1) and lateral prefrontal cortex are engaged in motor learning depending on the task [7,30]. A sequential finger-tapping task (FTT) is commonly used to evaluate motor-sequence learning by measuring the speed and accuracy of entering a presented number with the corresponding finger [31].
- Page 2, lines 80-81: authors should be more specific and indicate if anodal tDCS was applied to the ipsilateral motor cortex (as the acronym ‘iM1’ suggests).
- Response : To avoid confusion, we specified that anodal tDCS was applied to the M1 of the non-dominant hemisphere and that training was performed with the non-dominant hand. (Page 3, line 3-4)
- Regarding the figures: I suggest adding a figure that displays stimulation sites’ position on the scalp.
- Response : Thanks for your suggestion. A figure showing the location of the stimulus was added. (Figure 2, Page 3, Line 26)
- Page 8-9, Bibliography: authors should provide the abbreviated journal name in italics to the following citations Hummel et al., 2006; Elsner et al., 2019; Elsner et al., 2017; Antal et al., 2004; Keele et al., 2003; Doyon et al., 2018; Saucedo Marquez et al., 2013; Rahman et al., 2013
- Response : The abbreviated journal names has been added in italics for all references, including those pointed out. (Page 8-12)
- Page 9, lines 326-327: correct the authors’ name and order of citation to the following citations ‘CAUMO, W.; Souza, I.C.; Torres, I.L.; Medeiros, L.; Souza, A.; Deitos, A.; Vidor, L.; Fregni, F.; Volz, M.S.J.F.i.p. Neurobiological effects of transcranial direct current stimulation: a review. 2012, 3, 110. DOI:10.3389/fpsyt.2012.00110’.
- Response : There was an error in its endnote source. The author's name and order have been corrected as in the original. (Page 11, ref. No.39)

Reviewer 2 Report
The authors evaluated the timing effects of tDCS on motor performance in healthy individuals. Their main finding was that tDCS would be more effective on motor performance if given before the motor training. Although the topic is of great interest to the field, the study has a major flaw, described below, which limits the significance of the results substantially.
Major:
- Although the authors applied sham stimulation in the study design, it was only applied in one of the conditions, i.e., during finger-tapping, but not before finger-tapping. How can the authors know that their results before finger-tapping are real and not because of placebo effects?
Minor:
- The authors state in the introduction that previous studies have shown contradictory findings. Could the effects be age-dependent, as the authors suggest that for example with stroke there has been contradictory findings and stroke is more common in the elderly. The authors have mainly studied young individuals.
- The sham protocol has not been described. What was defined as sham?
Author Response
[Reviewer 2]
Comments and Suggestions for Authors
- The authors evaluated the timing effects of tDCS on motor performance in healthy individuals. Their main finding was that tDCS would be more effective on motor performance if given before the motor training. Although the topic is of great interest to the field, the study has a major flaw, described below, which limits the significance of the results substantially.
Major:
- Although the authors applied sham stimulation in the study design, it was only applied in one of the conditions, i.e., during finger-tapping, but not before finger-tapping. How can the authors know that their results before finger-tapping are real and not because of placebo effects?
- Response : Thanks for your important comments. In particular, we tried to compare the effect of tDCS between stimulation during training (CON group) and before training (PRI group), and to check the effect of only finger training through the sham group. Our findings showed better outcomes in PRI group than CON group, suggest the possibility that tDCS before training may be more effective than concurrent stimulation. Although the findings of our study will not be able to establish any generally accepted definitive conclusions, we believe that it may offer a partial contribution to this topic, and can encourage further research. Still, the shortcoming of the design you pointed out was acknowledged, so it was described as the first limitation (Page 8, line 15-18) and the conclusion sentence (Page 8, line 23-25) was revised to be more cautious.
- Revised first limitation (Page 8, line 15-18) : First, the blinding of participants may have been limited by the different treatment durations between the groups and the sham stimulation was only applied during training and not before training.
- Revised conclusion (Page 8, line 23-25) : Our findings suggest that the time shortly before motor training could be optimal for tDCS application. Further research is required to confirm these findings.
- The authors state in the introduction that previous studies have shown contradictory findings. Could the effects be age-dependent, as the authors suggest that for example with stroke there has been contradictory findings and stroke is more common in the elderly. The authors have mainly studied young individuals.
- Response : We agree that the effects of tDCS may differ among age groups, and the optimal timing may also differ for age and has not yet been established. We first tried to find the difference in healthy people. In three previous studies mentioned in introduction, the subjects of one study were a young group [1], the other two were the stroke subjects [2-3]. It is known that motor skill learning is promoted especially in young adults [4-6]. As relevant research, including our findings, accumulates, analysis on the timing-dependent effect according to age may be needed.
- Cabral, Maria E., et al. "Transcranial direct current stimulation: before, during, or after motor training?." Neuroreport11 (2015): 618-622.
- Jin, Minxia, et al. "Timing-Dependent interaction effects of tDCS with mirror therapy on upper extremity motor recovery in patients with chronic stroke: a randomized controlled pilot study." Journal of the neurological sciences 405 (2019): 116436.
- Liao, Wan-wen, et al. "Timing-dependent effects of transcranial direct current stimulation with mirror therapy on daily function and motor control in chronic stroke: a randomized controlled pilot study." Journal of NeuroEngineering and Rehabilitation1 (2020): 1-11.
- Hsu, Wan-Yu, et al. "Effects of noninvasive brain stimulation on cognitive function in healthy aging and Alzheimer's disease: a systematic review and meta-analysis." Neurobiology of aging8 (2015): 2348-2359.
- Perceval, Garon, Agnes Flöel, and Marcus Meinzer. "Can transcranial direct current stimulation counteract age-associated functional impairment?." Neuroscience & Biobehavioral Reviews 65 (2016): 157-172.
- Tatti, Elisa, et al. "Non-invasive brain stimulation of the aging brain: State of the art and future perspectives." Ageing research reviews 29 (2016): 66-89.
- The sham protocol has not been described. What was defined as sham?
- Response : The sham condition was further described. (Page 3, line 24-25)
- Page 3, line 24-25 : During the sham condition current flowed for a period of 30 s at the beginning of stimulation and then turned off [34].
- 34. Turi, Z.; Csifcsák, G.; Boayue, N.M.; Aslaksen, P.; Antal, A.; Paulus, W.; Groot, J.; Hawkins, G.E.; Forstmann, B.; Opitz, A.J.E.J.o.N. Blinding is compromised for transcranial direct current stimulation at 1 mA for 20 min in young healthy adults. Eur J Neurosci. 2019, 50, 3261-3268, DOI:10.1111/ejn.14403.

Round 2
Reviewer 2 Report
No further comments